# Conventional Structural Magnetic Resonance Imaging in Differentiating Chronic Disorders of Consciousness

**DOI:** 10.3390/brainsci8080144

**Published:** 2018-08-05

**Authors:** Sofya Morozova, Elena Kremneva, Dmitry Sergeev, Dmitry Sinitsyn, Lyudmila Legostaeva, Elizaveta Iazeva, Marina Krotenkova, Yulia Ryabinkina, Natalia Suponeva, Michael Piradov

**Affiliations:** Research Center of Neurology, 125367 Moscow, Russia; kremneva@neurology.ru (E.K.); dmsergeev@yandex.ru (D.S.); d_sinitsyn@mail.ru (D.S.); milalegostaeva@gmail.com (L.L.); lizaveta.mochalova@gmail.com (E.I.); krotenkova_mrt@mail.ru (M.K.); ryabinkina11@mail.ru (Y.R.); nasu2709@mail.ru (N.S.); mpi711@gmail.com (M.P.)

**Keywords:** conventional MRI, structural assessment, chronic disorders of consciousness, UWS, MCS, differential diagnosis

## Abstract

Differential diagnosis of unresponsive wakefulness syndrome (UWS) and minimally conscious state (MCS) is one of the most challenging problems for specialists who deal with chronic disorders of consciousness (DOC). The aim of the current study was to develop a conventional MRI-based scale and to evaluate its role in distinguishing chronic disorders of consciousness (Disorders of Consciousness MRI-based Distinguishing Scale, DOC-MRIDS). Data were acquired from 30 patients with clinically diagnosed chronic disorders of consciousness. All patients underwent conventional MRI using a Siemens Verio 3.0 T scanner, which included T2 and T1 sequences for patient assessment. Diffuse cortical atrophy, ventricular enlargement, sulcal widening, leukoaraiosis, brainstem and/or thalamus degeneration, corpus callosum degeneration, and corpus callosum lesions were assessed according to DOC-MRIDS criteria, with a total score calculation. The ROC-analysis showed that a reasonable threshold DOC-MRIDS total score was 5.5, that is, patients with DOC-MRIDS total score of 6 and above were classified as UWS and 5 and below as MCS, with sensitivity of 82.4% and specificity of 92.3%. The novel structural MRI-based scale for the assessment of typical brain lesions in patients with chronic DOC is relatively easy to apply, and provides good specificity and sensitivity values for discrimination between UWS and MCS.

## 1. Introduction

In clinical practice, the term “consciousness” has two dimensions: wakefulness and awareness [1,2]. Lack of awareness in patients who have regained arousal after coma is usually the result of extensive damage of several awareness-related structures [3]. As these patients maintain complete or partial preservation of hypothalamic and brainstem autonomic function (e.g., and circulation and spontaneous breathing, such condition, that is, chronic disorders of consciousness (DOC) may remain stable for a prolonged period [4]. Chronic DOC include the vegetative state (or unresponsive wakefulness syndrome, UWS) and minimally conscious state (MCS) [4,5,6].

According to the Multi-Society Task Force definition, unresponsive wakefulness syndrome is a clinical condition of complete unawareness of the self and the environment, with preserved sleep-wake cycles and hypothalamic and brainstem autonomic functions [7]. In contrast, minimally conscious state patients are able to demonstrate minimal but definite and reproducible behavioural evidence of self or environmental awareness [6].

Differential diagnosis of UWS and MCS is one of the most challenging problems for specialists who deal with chronic disorders of consciousness. Clinical examination remains a “gold standard” for detection of signs of consciousness, and therefore, for diagnosis [8]. However, behavioural assessment is often complicated due to severe motor deficit, aphasia, or mechanical barriers, such as a tracheostomy tube, fluctuating arousal levels, or ambiguous and rapidly habituating responses [9]. The rate of diagnostic errors may reach 40% [10]. The detection of signs of consciousness is crucial not only for routine management, particularly for the treatment of pain, but also for prognosis, which is significantly more favourable for MCS patients [6].

The Coma Recovery Scale-Revised (CRS-R) is the most commonly accepted clinical tool for the structured assessment of patients with chronic DOC; it yields optimal results (i.e., demonstrates excellent content validity, substantial internal consistency, and good interrater reliability) for differentiation between UWS and MCS, and may be used to assess chronic DOC with minor reservations, according to the report of the American Congress of Rehabilitation Medicine [11,12]. However, it has been observed that in a certain population of patients, the capacity for consciousness remains undetectable, even by flawlessly performed behavioural testing, and covert cognition, i.e., signs of wilful brain activity may be revealed in a small percentage of patients that respond mentally to active neuroimaging or electrophysiological paradigms, despite the lack of any behavioral response [13,14,15]. Therefore, multiple studies have been aimed at optimising chronic DOC differential diagnosis and outcome prediction using several neuroimaging techniques, for example, cerebral ^18^F-FDG PET, which was shown to complement bedside examinations and predict long-term recovery of patients with UWS [16]. Monti et al. demonstrated with fMRI that a small proportion of patients with DOC had brain activation while performing some mental-imagery tasks, suggesting some awareness and cognition [15]. Another study demonstrated an association between restoration of thalamo-frontal connectivity based on resting state fMRI data and the recovery of cognitive function [17]. Furthermore, functional connectivity analysis in another resting state fMRI study [18] showed that the default mode network, frontoparietal, salience, auditory, sensorimotor, and visual networks had a high discriminative capacity (>80%) for separating patients with MCS and UWS. Among them, the auditory network was ranked most highly, with bilateral auditory and visual cortices more functionally connected in patients with MCS compared to patients with UWS [18].

However, functional visualisation approaches are not widely available, require sophisticated data processing, and sometimes report inconsistent results, while structural imaging is a far more common method performed to estimate the extent of brain damage in patients with DOC. On a large sample of 143 patients with DOC, it was shown that prolonged disorders of consciousness were characterised by an extensive atrophy of subcortical structures, and clinical measures of awareness and wakefulness were negatively associated with the degree of tissue atrophy within the thalamic and basal ganglia nuclei, respectively [19]. Several attempts have been made to find patterns of brain damage attributable to UWS patients versus MCS. Significant atrophy of the left putamen and globus pallidus was shown in patients with UWS compared to the patients emerging from MCS [19]. There are data showing a decrease of thalamic volume in patients with UWS in comparison with MCS [20]. Moreover, when contrasting patients with UWS and MCS using an inclusive spatial default mode network mask at the group level, and without correcting for multiple comparisons, more prominent injury was detected in the ventromedial prefrontal cortex and the posterior cingulate cortex/praecuneus in those with UWS. Comparing patients in MCS− with those in MCS+, it was found that the latter had a more preserved left cerebral cortex, including the middle temporal gyrus, superior temporal gyrus, and inferior frontal gyrus (Broca’s area) [21]. There is evidence that some patterns of lesions may also have predictive value. A study of post-traumatic patients with UWS using conventional MRI data showed that patients who remained in UWS had a significantly higher frequency of corpus callosum, corona radiata, and dorsolateral brainstem injuries than did patients who recovered. Logistic regression analysis showed that corpus callosum and dorsolateral brainstem injuries were predictive of non-recovery [22]. In a study utilizing brain volumetry UWS and MCS, patients were discriminated by differences in the volumes of grey matter of the paracentral, para-hippocampal, inferior parietal, entorhinal, medial orbitofrontal cortex (i.e., areas belonging to the default mode network), and thalamus and caudate nucleus, as well as by the volumes of white matter regions involved in long range connectivity [23]. 

However, sophisticated approaches to visualization data analysis may not be available in the majority of clinical centres, and subjective behavioural assessment remains the mainstay of chronic DOC diagnosis. Considering the evidence of correlation between structural abnormalities and clinical scores [19], and the high availability of conventional MRI, we attempted to review several of the most common patterns of brain lesions in patients with DOC, combined within a rating scale, and to explore their association with clinical signs of awareness. This approach may help in establishing more accurate diagnosis in chronic DOC in a clinical setting using widely available conventional structural MRI data. The aim of the current study was to develop a conventional MRI-based scale that included the most common and clinically important structural damage found in patients with DOC, and to evaluate its role in distinguishing between the types of chronic disorders of consciousness (Disorders of Consciousness MRI-based Distinguishing Scale, DOC-MRIDS).

## 2. Materials and Methods

Data were acquired from 30 patients (17 males, 13 females) aged 21–61 years (median age 32, first and third quartiles (23, 50)) with clinically diagnosed chronic disorders of consciousness (13 with minimal consciousness state (MCS) and 17 with unresponsive wakefulness syndrome (UWS)) due to anoxia (including stroke patients with cardiac arrest) (20), trauma (9) and demyelination (1). The duration of DOC in all participants was at least 5 months from the incident. Three experts with at least 3 years of experience in management of chronic DOC performed CRS-R assessment in each patient independently, and the decision was made in favour of the highest consciousness state measured across the assessments. 

Informed consent was obtained from all official representatives of individual participants included in the study. All procedures performed in this study involving human participants were in accordance with the ethical standards of the institutional and/or national research committee and with the 1964 Declaration of Helsinki and its later amendments or comparable ethical standards, and the protocol and Informed Consent Form was approved by the Ethics Committee of Research Center of Neurology (project identification number 11/14). The Institutional Review Board has also approved the undertaking of this study in accordance with local regulations.

All patients underwent MR examination using a Siemens Verio 3.0 T scanner; the scanning protocol included conventional spin echo T2 sequence (TR 4000 ms; TE 118 ms; thickness 5.0 mm; spacing 1.5 mm; duration: 2 min 02 s) and three-dimensional gradient echo T1 sequence (TR 1900 ms, TE 2.5 ms; thickness 1.0 mm; spacing 1.0 mm; slices 176; duration: 4 min 18 s), which were used for patient assessment. 

The following parameters were suggested for DOC-MRIDS: diffuse cortical atrophy, ventricular enlargement, sulcal widening, leukoaraiosis, brainstem and/or thalamus degeneration, corpus callosum degeneration and corpus callosum lesions (Table 1). These structural pathologies were chosen for assessment as they were the most frequently revealed in previous studies [22,24]. 

For diffuse cortical atrophy, brainstem and/or thalamus degeneration and corpus callosum lesions, the presence (1) or absence (0) of the pathology was taken into consideration. The presence of diffuse cortical atrophy and brainstem and/or thalamus degeneration was reported in cases of undoubted prominent volume loss of the above mentioned structures, that is, obvious widespread thinning of the cerebral cortex and unquestionable bilateral and symmetrical decrease in size of the thalamus and/or brainstem structures. Unilateral damage due to stroke or traumatic injury was not taken into consideration. The presence of a corpus callosum lesion was reported in cases of hypointense regions on sagittal T1 images, indicating white matter destruction in this area. 

Other parameters were assessed as absent (0), moderate (1), or severe (2). Moderate ventricular enlargement was characterised by the Evans index and ranged from 0.31 to 0.74, while severe ventricular enlargement was defined by Evans index >0.74 [25]. Enlargement of subarachnoid spaces up to 0.4 cm was classified as moderate sulcal widening, while dilatation >0.4 cm was considered as severe [26]. Periventricular caps, smooth halos, and diffuse periventricular areas of intensity changes denoted moderate leukoaraiosis, while extensive confluent T2-hyperintensive areas spreading to deep and even subcortical white matter were rated as severe leukoaraiosis. Corpus callosum degeneration was assessed by the thickness of its central part: 0.4–0.2 cm indicated moderate and <0.2 cm severe degeneration [27]. The total score ranged from zero (normal brain structure) to 11 points (extensive brain damage). The scoring criteria are visualized on Figure 1.

T2-weighted images (WI) and T1WI of all patients were assessed according to the scale resulting in a total score. Images were analysed by two neuroradiology experts who were unaware of the diagnosis. Statistical analysis was performed using SPSS Statistics 20 (IBM Corp., New York, NY, USA).

## 3. Results

The DOC-MRIDS total score in the studied patients ranged from 1 to 11. The CRS-R and DOC-MRIDS detailed and total scores for each patient are presented in Table 2. Age, gender, and aetiology did not differ between MCS and UWS groups (Mann-Whitney U Test, *p* > 0.05). The conscious state (MCS) was taken as the default condition for the binary classification test, which means that sensitivity scores were calculated on the unresponsive (UWS) patients, while specificity scores were calculated on the conscious (MCS) patients. ROC-analysis showed that a reasonable threshold score was 5.5, which means that patients with a DOC-MRIDS total score of ≥6 were classified as UWS (Figure 2), while the score of ≤5 indicated MCS (Figure 3), with sensitivity of 82.4% and specificity 92.3% (area under the curve (AUC) = 90.5%, *p* = 0.000056, Figure 4). Because it is more diagnostically important not to miss patients in MCS, one can increase specificity to 100% by setting a threshold score of 7.5, although there is then a sensitivity drop to 52.9%. In our sample, 100% of patients with a total score of ≥8 were clinically diagnosed with UWS.

The Mann-Whitney U Test did not reveal significant differences of DOC-MRIDS total scores between patients with DOC due to traumatic brain injury and DOC of non-traumatic origin both in UWS and in MCS groups (*p* > 0.05).

Analysis of the estimated parameter prevalence showed that the majority of UWS patients had diffuse cortical atrophy (70.6%) and brainstem and/or thalamus degeneration (88.2%), and only this group was characterised by severe ventricular enlargement and corpus callosum degeneration. Thalamus degeneration coincided with brainstem degeneration in all but 3 patients without thalamus degeneration, all of whom were diagnosed as UWS. Isolated thalamus degeneration was not revealed in our sample of patients. At the same time, the majority of MCS patients were characterised by absence of leukoaraiosis (69.2%), brainstem and/or thalamus degeneration (69.2%), corpus callosum degeneration (77.0%), and diffuse cortical atrophy (85.0%) (Table 3.).

## 4. Discussion

We proposed a novel MRI-based scale for distinguishing the types of chronic disorders of consciousness, that is, UWS and MCS; it is based on the rating of the presence and severity of the commonly found structural abnormalities in these categories of patients. Taking into consideration the obtained relatively high sensitivity and specificity, DOC-MRIDS may be used as an additional tool for behavioural assessment in discriminating between types of chronic disorders of consciousness in stable condition.

The undoubted advantage of the DOC-MRIDS is its availability, since it requires only conventional MRI data for the assessment, which is easily accessible on all MRI scanners. Another beneficial feature is that motion artefacts, which pose a common problem for advanced MRI techniques and require medical sedation, can be easily reduced in conventional MRI techniques by the standard reconstruction algorithm available at the majority of scanners (GE (*PROPELLER*), Siemens (*BLADE*), Philips (*MultiVane*), Hitachi (*RADAR*), and Cannon (*JET*)), providing images accurate enough for assessment with the DOC-MRIDS. 

To our knowledge, this study is the first in which a wide range of pathological morphological features have been combined within a single scale for assessment of patients with chronic disorders of consciousness due to different causes. However, the data obtained are in agreement with some previous studies on patients with early posttraumatic brain injuries causing disorders of consciousness. Namely, it has been shown that corpus callosum and dorsolateral brainstem injuries were predictive of non-recovery, and a lesion volume exceeding 4 mL in the corpus callosum or 1.5 mL in the brainstem was likely to lead to an unfavourable outcome [22,23]. The majority of our patients with UWS also had brainstem degeneration and severe corpus callosum degeneration, unlike the patients with MCS, CRS-R domains, and total score also negatively correlated with diffuse cortical atrophy, leukoaraiosis, brainstem and/or thalamus degeneration and corpus callosum degeneration. These data give evidence of the undoubted importance of the morphological integrity of these structures for awareness.

The results of the scale assessment were applicable to patients with DOC independently of traumatic versus non-traumatic aetiology. Advanced MRI-based analyses of brain matter abnormalities in chronic DOC demonstrated that non-traumatic injuries might be more widespread, while the location of traumatic lesions is somewhat more specific, and primarily involves the brainstem, thalamus, and corpus callosum [19]. However, at the selected level of structural MRI assessment, no difference between aetiology-based cohorts of patients was established.

Yet, this study has a number of limitations. As the scale was intended to be used only in patients with DOC, it was tested on a cohort of patients with established chronic DOC of various aetiologies and clinical severity without using a control group of individuals with preserved consciousness. The MRI parameters taken into consideration are common for various medical conditions that are not associated with any disorder of consciousness (i.e., leukoaraiosis in small vessel disease, ventricular enlargement in hydrocephalus, corpus callosum lesions in multiple sclerosis, sulcal widening in degenerative brain diseases, among others), and comparing the results of the scale assessment among these categories of patients with the UWS or MCS groups seemed to be incorrect. We also did not apply the scale to coma patients, as it is an acute condition with unstable clinical and radiological features. Besides the absence of a control group, an important shortcoming is the limited number of patients, due to the pilot nature of the study. The results obtained should be verified on a larger population. The heterogeneity of the sample concerning age of patients and aetiology of brain injury that is often seen in this clinical setting should be also noted. The second disadvantage is that the assessment may be operator-dependent. Indeed, the evaluation of some parameters could be quite subjective, and requires the evaluator to be experienced in neuro-MRI interpretation. Another serious limitation is that behavioural tests that are considered as standards for the clinical assessment of consciousness do not provide absolute accuracy, even when performed with the use of CRS-R. This problem could partly explain the fact that, in spite of the relatively high correlation of DOC-MRIDS with CRS-R, there are outliers in the UWS group with almost “normal” brain on MRI (Figure 5). Maybe, some of these outlying results might indicate patients with a discrepancy between absent behavioural signs of consciousness and the presence of some brain activity, suggesting potential for at least partially preserved consciousness [28,29]. However, advanced MRI and neurophysiological techniques are required to detect such an activity. 

## 5. Conclusions

Integrated assessment of the structural abnormalities typical for patients with chronic DOC with the proposed novel conventional MRI-based scale helps to establish the clinical type of DOC with considerable specificity and sensitivity, and may be used as a complement to clinical assessment. However, more patient data are needed to confirm the findings and to estimate the prognostic value of the scale.

## Figures and Tables

**Figure 1 brainsci-08-00144-f001:**
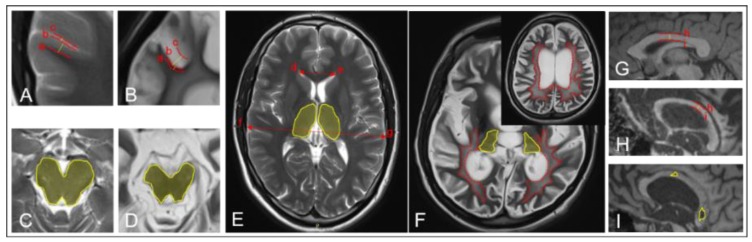
Visualization of scoring parameters on healthy (**A**,**C**,**E**,**G**) and damaged (**B**,**D**,**F**,**H**,**I**) brain: a-b distance (yellow line between a and b on (**A**,**B**))—cortical thickness; b-c distance (yellow line between b and c on (**A**,**B**))—sulcal width; yellow marked areas on (**C**,**D**)—preserved (**C**) and degenerated (**D**) brainstem; yellow marked areas on (**E**,**F**)—preserved (**E**) and degenerated (**F**) thalamus; d-e distance to f-g distance ratio on (**E**)—Evans index calculation for ventricular enlargement measuring; red dotted line marked areas on (**F**)—confluent areas of leukoaraiosis; h-i distance (yellow line between h and i on (**G**,**H**))—thickness of corpus callosum central part; yellow marked areas on (**I**)—hypointense corpus callosum lesions.

**Figure 2 brainsci-08-00144-f002:**
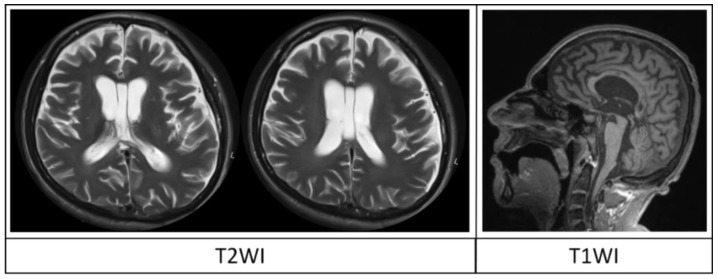
T2-WI and T1-WI of patient G., male, aged 26, in MCS due to traumatic brain injury 6 months prior to MRI (see patient No 10 in Table 2 for detailed information). Conventional MRI revealed ventricular enlargement with Evans index of 0.34 (DOC-MRIDS score 1), sulcal widening up to 0.3 cm (score 1) and corpus callosum lesions (score 1), without diffuse cortical atrophy, leukoaraiosis, corpus callosum and brainstem degeneration (score 0). The total DOC-MRIDS score was 3, corresponding to MCS, and was in agreement with clinical examination (CRS-R-11).

**Figure 3 brainsci-08-00144-f003:**
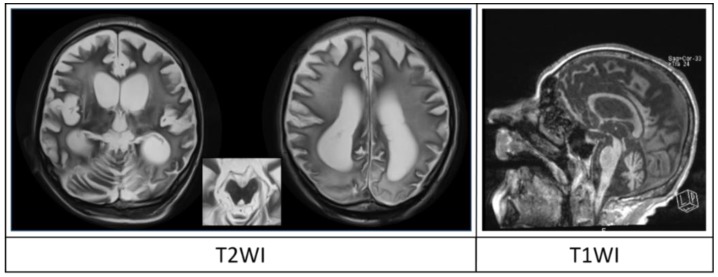
T2-WI and T1-WI of patient O., male, aged 56, in UWS due to an episode of anoxia 5 months prior to MRI (see patient No 26 in Table 2 for detailed information). Conventional MRI demonstrated diffuse cortical atrophy (score 1), moderate ventricular enlargement with Evans index of 0.41 (score 1) and severe sulcal widening up to 0.7 cm (score 2); severe leukoaraiosis with confluent diffuse periventricular areas of changed intensity spreading to the deep white matter and up to the very cortex (score 2), moderate corpus callosum degeneration with the thickness of its central part at 0.3 cm (score 1) and brainstem and thalamus degeneration (score 1). The total DOC-MRIDS score is 8, which corresponds to UWS (CRS-R-6, also corresponding to UWS).

**Figure 4 brainsci-08-00144-f004:**
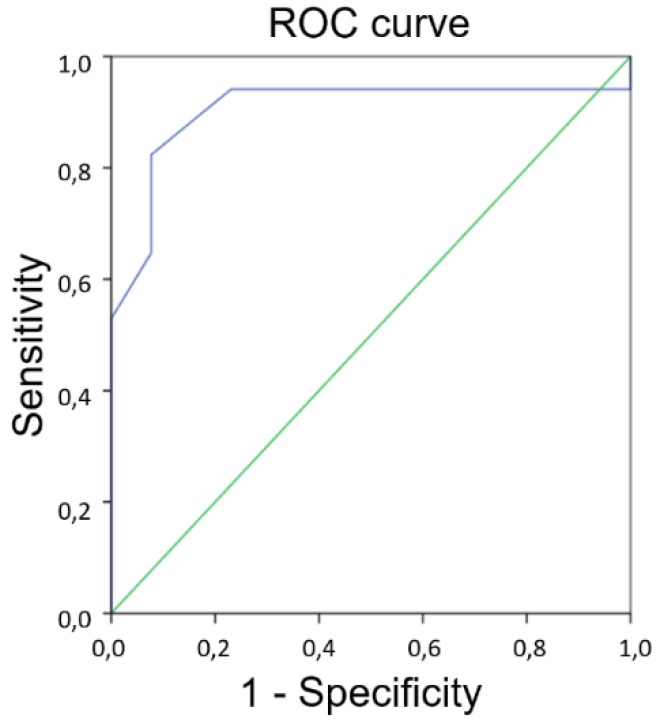
ROC curve for discrimination between UWS and MCS patients. AUC = 90.5%, *p* = 0.000056. Spearman correlation analysis showed significant negative correlation between CRS-R score and DOC-MRIDS score with Spearman ρ = −0.68, *p* = 0.01 (Figure 5). There also were significant (*p* < 0.05) negative correlations between CRS-R score and diffuse cortical atrophy (Spearman ρ = −0.6), leukoaraiosis (Spearman ρ = −0.62), brainstem and/or thalamus degeneration (Spearman ρ = −0.51) and corpus callosum degeneration (Spearman ρ = −0.58). CRS-R auditory, visual, motor and communication domains significantly (*p* < 0.05) correlated with the same structural parameters (Spearman ρ ranged from−0.5 to −0.65).

**Figure 5 brainsci-08-00144-f005:**
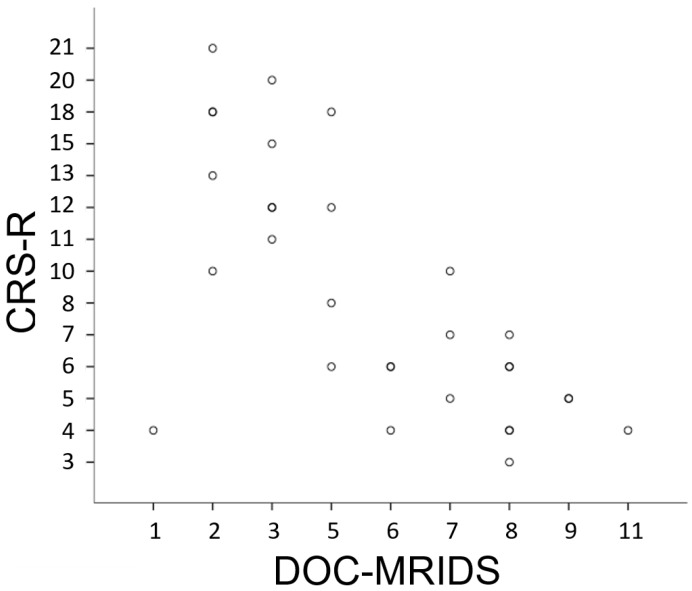
Inverse rank correlation between CRS-R and DOC-MRIDS scores. Spearman ρ = −0.68, *p* = 0.01.

**Table 1 brainsci-08-00144-t001:** MRI parameters for patient assessment and corresponding scores.

Parameters	Score			
	Absence	Presence	Moderate Changes Presence	Severe Changes Presence
diffuse cortical atrophy	0	1	-	-
brainstem and/or thalamus degeneration	0	1	-	-
corpus callosum lesions	0	1	-	-
ventricular enlargement	0	-	1	2
sulcal widening	0	-	1	2
leukoaraiosis	0	-	1	2
corpus callosum degeneration	0	-	1	2

**Table 2 brainsci-08-00144-t002:** Demographic data, CRS-R and DOC-MRIDS total and detailed scores for each patient.

No.	Age	Sex	Aetiol *	CRS-R	Aud	Vis	Mot	Ver	Comm	Arous	DOC	DOC-MRIDS	DCA	VE	SW	Leu	BTD	CCD	CCL
1	23	M	TR	20	4	5	5	2	1	3	MCS	3	0	1	1	0	0	0	1
2	50	M	ANOX	10	2	1	3	1	1	2	MCS	2	0	1	0	0	0	0	1
3	21	M	ANOX	4	0	0	1	1	0	2	UWS	8	1	2	0	2	1	2	0
4	31	F	ANOX	7	1	1	2	1	0	2	UWS	8	1	2	1	2	1	1	0
5	61	F	ANOX	5	0	0	2	1	0	2	UWS	9	1	1	1	2	1	2	1
6	31	F	ANOX	18	4	4	5	1	2	2	MCS	2	0	1	0	0	0	0	1
7	49	F	TR	6	1	0	2	1	0	2	UWS	7	0	2	0	1	1	2	1
8	22	M	ANOX	6	1	0	2	1	0	2	UWS	8	1	2	1	2	1	1	0
9	19	F	TR	6	1	0	2	1	0	2	UWS	5	0	2	0	1	1	1	0
10	26	M	TR	11	2	3	3	1	0	2	MCS	3	0	1	1	0	0	0	1
11	23	M	TR	8	1	1	2	2	0	2	UWS	5	0	1	0	1	1	1	1
12	22	M	TR	3	0	0	1	1	0	1	UWS	8	1	2	0	2	1	2	0
13	50	M	ANOX	6	1	0	2	1	0	2	UWS	6	1	1	2	1	0	1	0
14	24	F	TR	21	4	5	5	1	2	3	MCS	2	0	1	0	1	0	0	0
15	55	M	ANOX	15	2	2	4	2	2	3	MCS	3	0	1	1	1	0	0	0
16	55	M	ANOX	4	0	1	1	1	0	1	UWS	6	1	1	1	2	1	0	0
17	56	F	DEM	18	3	4	4	2	2	3	MCS	5	0	1	1	0	1	1	1
18	22	F	TR	7	1	1	2	1	0	2	UWS	7	1	2	0	1	1	1	1
19	32	M	ANOX	18	4	4	5	1	2	2	MCS	2	0	1	0	0	0	1	0
20	28	F	ANOX	4	0	0	1	1	0	2	UWS	1	0	0	1	0	0	0	0
21	51	M	ANOX	5	1	0	1	1	0	2	UWS	9	1	1	2	2	1	2	0
22	24	F	ANOX	4	0	0	1	1	0	2	UWS	8	1	1	2	2	1	1	0
23	33	M	ANOX	12	1	2	3	2	1	3	MCS	3	0	1	0	0	1	0	1
24	53	M	ANOX	12	2	3	3	1	1	2	MCS	5	1	1	0	2	1	0	0
25	43	F	ANOX	13	3	3	3	1	1	2	MCS	2	0	1	1	0	0	0	0
26	56	M	ANOX	6	1	0	2	1	0	2	UWS	8	1	1	2	2	1	1	0
27	46	F	ANOX	6	1	0	2	1	0	2	UWS	11	1	2	2	2	1	2	1
28	47	M	ANOX	10	2	3	2	1	0	2	MCS	7	1	1	2	2	0	1	0
29	22	M	TR	12	3	3	3	1	0	2	MCS	3	0	1	0	0	1	0	1
30	47	M	ANOX	6	1	0	2	1	0	2	UWS	6	0	1	1	1	1	1	1

*—aetiology: TR—trauma, ANOX—anoxia (including stroke patients with cardiac arrest), DEM—demyelination. Sex: M—male, F—female. CRS-R domains: aud—auditory; vis—visual; mot—motor; ver—verbal; comm—communication; arous—arousal. DOC—types of chronic disorders of consciousness. DOC-MRIDS parameters: DCA—diffuse cortical atrophy; VE—ventricular enlargement; SW—sulcal widening; leu—leukoaraiosis; BTD—brainstem and/or thalamus degeneration; CCD—corpus callosum degeneration; CCL—corpus callosum lesion.

**Table 3 brainsci-08-00144-t003:** Prevalence of DOC-MRIDS-assessed parameters in UWS and MCS patients.

Parameters	Prevalence in UWS/MCS Patients, %			
	Absent	Present	Moderate Changes	Severe Changes
diffuse cortical atrophy	29.4/85.0	70.6/15.0	-	-
brainstem and/or thalamus degeneration	11.8/69.2	88.2/30.8	-	-
corpus callosum lesions	64.7/46.2	35.3/53.8	-	-
ventricular enlargement	5.9/0.0	-	47.0/100	47.0/0.0
sulcal widening	35.3/53.8	-	35.3/38.5	29.4/7.7
leukoaraiosis	5.9/69.2	-	35.3/15.4	58.8/15.4
corpus callosum degeneration	11.8/77.0	-	52.9/23.0	35.3/0.0

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
