# Peer review of "Conventional Structural Magnetic Resonance Imaging in Differentiating Chronic Disorders of Consciousness"

_brainsci, 2018, doi:10.3390/brainsci8080144_

Round 1

Reviewer 1 Report

Major comments:

The introduction cites several papers regarding structural MRI findings in UWS and MCS patients. However, literature regarding acute and chronic patients is mixed while I think the introduction would improve by making a clear distinction.

line 91-91 describes that the latest structural MRI techniques were not reliable to differentiate between UWS and MCS. The authors have not reviewed Annen et al., (2018) where a completely automated approach for the classification of UWS and MCS patients was employed with a high accuracy.

A group of 30 patients is very small to establish the accuracy of a newly proposed rating scale, and is a big limitation of the current study

The CRS-R has been performed only 3 times, while there is evidence that some clinical misdiagnosis could occur if the CRS-R has not been performed 5 times (see Wannez et al.,). Clinical misdiagnosis (with the CRS-R) is mentioned as limitation in the discussion, in the context of UWS patients with a high DOC-MRIDS score. Do the authors have any data regarding outcome of those patients?

What is the rationale behind the choice for only considering bilateral damage? Especially in TBI patients the brain damage is often unilateral.

The scoring criteria should be explained more clearly or visualized with example scans.

The neuro-radiologists were blinded to the aetiology. However, the cause of brain damage is clear from the structural MRI for any experienced radiologist. Therefore I think the raters cannot have been blinded to the aetiology. Following from this, it could be that the aetiology is a good predictor for diagnosis. The results section however mentions that aetiology did not differ per diagnosis, however, no statistics are provided to support this statement. The absence of a significant difference could also be due to the limited sample size.

The results section should provide group statistics regarding demographic information (does age/gender/aetiology differ between diagnoses).

In the discussion the inter-rater variability is mentioned as disadvantage. Indeed inter-rater variability is a problem when using rating scales. Could the authors provide the scores of both raters? How did the raters come to an agreement? Did the authors test for inter-rater variability using the kappa statistic?

Small comments:

In the abstract and introduction 1 is written as a number instead of word.

The citations 4 and 5 are not appropriate for referring to UWS and MCS.

UWS: Monti et al., (2010) + Laureys et al., (2010). MCS: Giancino et al. (2002).

MCS- and MCS+ are mentioned but not described enough. Also it's not of interest for the rest of the paper so consider to remove this from the introduction.

In which center was the data acquired?

Were the patients sedated for the MRI to reduce movement and improve MRI quality? The discussion section mentions several post-processing steps to reduce movement artefacts, but sedation is an option for structural MRI acquisition as well.

"responsive state" is used to refer to MCS several times, while it is a misleading term. Not all MCS patients give response to command.

Mention which co-authors were responsible for the CRS-R and for the MRI rating.

Was the informed consent written?

Table 2 is hard to read with only numbers. Readability of the table would improve if Gender (Sex), Aetiology and Diagnosis (DOC) would be mentioned by abbreviation directly.

Table 3: The present column is the direct inverse of the absent column and could therefore be removed.

line 219-225 of the discussion are hard to follow in their current location. Consider to move them to the limitation section.

line 242 "normal cognition" is a huge leap from the distinction between UWS and MCS, as MCS patients do not have near normal cognitive capacities. Consider to use "awareness" instead.

line 248 shows a strange formatting of the citation.

Author Response

Response to Reviewer 1 Comments:

Thank you for detailed review of our paper. We have answered each of your points below.

Point 1: The introduction cites several papers regarding structural MRI findings in UWS and MCS patients. However, literature regarding acute and chronic patients is mixed while I think the introduction would improve by making a clear distinction. 

Response 1: The literature review within the Introduction section was refined according to reviewer’s suggestions. Only data pertaining to chronic settings were included.

Point 2: line 91-91 describes that the latest structural MRI techniques were not reliable to differentiate between UWS and MCS. The authors have not reviewed Annen et al., (2018) where a completely automated approach for the classification of UWS and MCS patients was employed with a high accuracy. 

Response 2: Review of paper by Annen et al., (2018) has been added to the Introduction section and the remaining text has been revised.

Lines 90-94: In a study utilizing brain volumetry UWS and MCS patients were discriminated by differences in the volumes of grey matter of the paracentral, para-hippocampal, inferior parietal, entorhinal, medial orbitofrontal cortex (i.e., areas belonging to the default mode network), and thalamus and caudate nucleus, as well as by the volumes of white matter regions involved in long range connectivity [Annen et al., 2018].

Point 3: A group of 30 patients is very small to establish the accuracy of a newly proposed rating scale, and is a big limitation of the current study.

Response 3: The sample size is discussed in the limitations section, however, we considered it admissible due to the pilot nature of the study, further clarification of the utility of scale is being assessed in larger population.

Lines 309-311: Besides lack of a control group, an important shortcoming is the limited number of patients due to the pilot nature of the study. The results obtained should be verified on a larger population.

Point 4: The CRS-R has been performed only 3 times, while there is evidence that some clinical misdiagnosis could occur if the CRS-R has not been performed 5 times (see Wannez et al.,). Clinical misdiagnosis (with the CRS-R) is mentioned as limitation in the discussion, in the context of UWS patients with a high DOC-MRIDS score. Do the authors have any data regarding outcome of those patients?

Response 4:  Indeed, the risk of misdiagnosis in chronic DOC is lower when multiple assessments are performed. As the duration of DOC in subjects was at least 5 months, that is associated with lower chances for improvement, and there was no need for repeated assessment of behavioral changes as a result of some intervention (patients were evaluated only once during the study soon after the admission, no intervention was performed), we considered employing assessment with 3 independent raters with at least 3 years of experience in DOC management and familiar with the CRS-R as a sufficient measure to minimize the bias. Regarding UWS patients with high DOC-MRIDS score, we do not have any data about outcome yet as they are still in the same state. We should also note, that data collection for the present study has been started in our center since 2016 with already approved design and the mentioned study by Wannez et al. has been published in 2017, therefore we decided not to change our protocol to ensure the consistency of results.

Point 5: What is the rationale behind the choice for only considering bilateral damage? Especially in TBI patients the brain damage is often unilateral.

Response 5: The existence of a large number of stroke patients with unilateral damage and fully intact consciousness suggests that unilateral degeneration is less likely to cause DOC. Thus we limited the cortical atrophy, brainstem and/or thalamus degeneration subscales to bilateral damage.

Point 6: The scoring criteria should be explained more clearly or visualized with example scans.

Response 6: The scoring criteria have been visualized on the new figure included in Material and Methods section. (lines 154-164 of the manuscript, please, find Figure 1 in the attached file with Raters individual scores).

Point 7: The neuro-radiologists were blinded to the aetiology. However, the cause of brain damage is clear from the structural MRI for any experienced radiologist. Therefore, I think the raters cannot have been blinded to the aetiology. Following from this, it could be that the aetiology is a good predictor for diagnosis. The results section however mentions that aetiology did not differ per diagnosis, however, no statistics are provided to support this statement. The absence of a significant difference could also be due to the limited sample size.

Response 7: Statistic data supporting the statement that aetiology did not differ per diagnosis have been included in the results section. Phrase that raters were blinded to etiology has been removed (Line 167).

Lines 229-231: The Mann-Whitney U Test did not reveal significant differences of DOC-MRIDS total scores between patients with DOC due to traumatic brain injury and DOC of non-traumatic origin both in UWS and in MCS groups (p>0,05).

Point 8: The results section should provide group statistics regarding demographic information (does age/gender/aetiology differ between diagnoses).

Response 8: Group statistics regarding demographic information have been included in the Results section.

Lines 171-172: Age, gender and aetiology didn’t differ between MCS and UWS groups (Mann-Whitney U Test, p>0,05).

Point 9: In the discussion the inter-rater variability is mentioned as disadvantage. Indeed inter-rater variability is a problem when using rating scales. Could the authors provide the scores of both raters? How did the raters come to an agreement? Did the authors test for inter-rater variability using the kappa statistic?

Response 9: The scores of the both raters (SM and EK) are in the attachment, however there were no principal discrepancies between the raters, therefore Cohen’s kappa for MCS/UWS classification based on the total DOC-MRIDS score was equal to 1. There were some discrepancies in the parameters scores, agreement was obtained after a consilium with the third specially invited rater (MK).

Point 10: In the abstract and introduction 1 is written as a number instead of word. 

Response 10: Number has been changed to word.

Lines 11,42: Differential diagnosis of unresponsive wakefulness syndrome (UWS) and minimally conscious state (MCS) is one of the most challenging problems for specialists who deal with chronic disorders of consciousness (DOC).

Point 11: The citations 4 and 5 are not appropriate for referring to UWS and MCS. UWS: Monti et al., (2010) + Laureys et al., (2010). MCS: Giancino et al. (2002).

Response 11: The citations have been changed according to reviewer’s suggestion.

Point 12: MCS- and MCS+ are mentioned but not described enough. Also it's not of interest for the rest of the paper so consider to remove this from the introduction. 

Response 12: Sentence was deleted as suggested by the reviewer.

Point 13: In which center was the data acquired?

Response 13: All clinical and radiological data were acquired and processed at the Research Center of Neurology, (Moscow, Russia).

Point 14: Were the patients sedated for the MRI to reduce movement and improve MRI quality? The discussion section mentions several post-processing steps to reduce movement artefacts, but sedation is an option for structural MRI acquisition as well.

Response 14: In 4 patients (13,3%) an anesthesiologist induced light sedation by dexmedetomidine administration via intravenous infusion at a constant rate of 1 µg/kg/h, because scanning protocol for this study included also resting state functional MRI and diffusion-weighted MRI. During the infusion period, the anesthesiologist monitored cuff blood pressure, electrocardiogram and pulse oximetry. However, sedation complicates the examination and possibility of using only routine techniques is an advantage.

Point 15: "responsive state" is used to refer to MCS several times, while it is a misleading term. Not all MCS patients give response to command.

Response 15: “Responsive state” has been changed to “conscious state” as MCS patients by definition demonstrate signs of consciousness.

Lines 172-175: The conscious state (MCS) was taken as the default condition for the binary classification test, which means that sensitivity scores were calculated on the unresponsive (UWS) patients while specificity scores were calculated on the conscious (MCS) patients.

Point 16:  Mention which co-authors were responsible for the CRS-R and for the MRI rating.

Response 16: Information about co-authors responsible for MRI and CRS-R rating was included in the corresponding section.

Lines 330-335: Author Contributions: Conceptualization, S.M. and E.K.; Data Curation, S.M., E.K., L.L. and E.I.; Methodology, S.M. and E.K.; MRI rating, S.M. and E.K.; CRS-R rating L.L, E.I., D.Ser; Formal Analysis, S.M. and D.Syn.; Investigation, S.M., E.K., L.L., E.I., D.Ser., Yu.R.; Validation, E.K., D.Syn., D.Ser. and N.S.; Resources, N.S., Yu.R., and M.K.; Writing – Original Draft Preparation, S.M.; Writing – Review & Editing, D.Ser., D.Syn., N.S., M.P.; Supervision, N.S., M.K. and M.P.; Project Administration, M.P.; Funding Acquisition, M.P.

Point 17: Was the informed consent written?

Response 17: Informed consent was written and was signed by the official representatives of patients.

Point 18: Table 2 is hard to read with only numbers. Readability of the table would improve if Gender (Sex), Aetiology and Diagnosis (DOC) would be mentioned by abbreviation directly.

Response 18: Table 2 has been changed according to the reviewer’s suggestions

Point 19: Table 3: The present column is the direct inverse of the absent column and could therefore be removed.

Response 19: We would like to preserve all columns in table 3 as it seems to be more demonstrative.

Point 20: line 219-225 of the discussion are hard to follow in their current location. Consider to move them to the limitation section.

Response 20: The corresponding text has been moved to limitation section according to reviewer’s suggestion (lines 302-311).

Point 21: line 242 "normal cognition" is a huge leap from the distinction between UWS and MCS, as MCS patients do not have near normal cognitive capacities. Consider to use "awareness" instead.

Response 21: "normal cognition" has been changed to “awareness” as suggested by reviewer.

Lines 294-295: These data give evidence of the undoubted importance of morphological integrity of these structures for awareness.

Point 22: line 248 shows a strange formatting of the citation.

Response 22: Formatting of the citation has been corrected.

Reviewer 2 Report

The authors had an aim that the current study was to develop a conventional MRI-based scale in distinguishing chronic disorders of consciousness. The structural MRI-based scale for the assessment of typical brain lesions in patients showed diffuse cortical atrophy, ventricular enlargement, sulcal widening, leukoaraiosis, brainstem and/or thalamus degeneration, corpus callosum degeneration and corpus callosum lesions. There were no critical concerns for the overall configuration, therefore this reviewer would like to raise some suggestions that can be more improved way.

1. There is no description of the fundamental purpose of studying unresponsive wakefulness syndrome based on MRI analysis. In order to clearly describe the purpose of the current study, it should provide a plausible basis for the superiority of the current research to existing methods.

2. There is missing some information in methodology for clinical practice. There is a lack of information on how to obtain the research consent of the patients and on the IRB approval information to conducting the study.

3. Despite providing a variety of clinical implications through MRI, there is a lack of providing correlation of data. It means that this reviewer suggests dividing the clinical evaluations presented in Table 2 into age or gender-related phenomena and preparing data to measure the correlation.

4. It is necessary to provide information on the patient number that images shown in Figures 3 and 4. And, when providing a representative images for disease, it is recommended to compare with the relatively normal images or non-induced images in your cohort.

5. In a somewhere paragraph of discussion, it should be described the reason of the various diseases, and how to gather the benefits of analyzing them by MRI, as compared to existing methods. And the limitations of this study must also be included.

Author Response

Please, find a point-by-point response in the attached file

Reviewer 3 Report

Lines 30-33: How dissociation between 2 components of consciousness results in the development of stable conditions need to be explained and evidenced with current literature.

Lines 48-50: ‘’Optimal results’’ of the Coma Recovery Scale-Revised (CRS-R) need to be elaborated on.

Lines 50-51: ‘’certain’’ population needs to be detailed.

Lines 54-56: Evidence cited does not relate to optimisation of chronic DOC differential diagnosis using neuroimaging. More robust review of the literature is needed to provide rationale for the study and also to review how MRI has already been used in DOC diagnosis.

Lines 91-100: Justification for the current study needs to be given. The current argument appears weak, with the paper highlighting that ‘’the latest structural MRI analysis techniques were not able to reliably differentiate between UWS and MCS at the single-subject level’’.

Lines 190-209: Examples of the 2 case studies should go in the footer of the MRI images rather than as a separate section.

Author Response

(The authors gave the same response as above.)

Round 2

Reviewer 1 Report

The authors have modified the manuscript according to most of the feedback, which I believe has improved the quality of the manuscript.

Reviewer 2 Report

All concerns that were raised by this reviewer have been well addressed. I agree to the publication with the present form.

Reviewer 3 Report

-